# Inhibition of intracellular lipolysis promotes human cancer cell adaptation to hypoxia

Xiaodong Zhang[1], Alicia M Saarinen[1,2], Taro Hitosugi[3], Zhenghe Wang[4], Liguo Wang[5], Thai H Ho[6], Jun Liu[1,2,7]*

[1]Department of Biochemistry and Molecular Biology, Mayo Clinic in Arizona, Scottsdale, United States; [2]HEALth Program, Mayo Clinic in Arizona, Scottsdale, United States; [3]Department of Pharmacology, Mayo Clinic, Rochester, United States; [4]Department of Genetics and Genome Sciences, Case Medical Center, Case Western Reserve University, Cleveland, United States; [5]Division of Biomedical Statistics and Informatics, Mayo Clinic, Rochester, United States; [6]Division of Hematology and Medical Oncology, Mayo Clinic in Arizona, Scottsdale, United States; [7]Division of Endocrinology, Mayo Clinic in Arizona, Scottsdale, United States

**Abstract** Tumor tissues are chronically exposed to hypoxia owing to aberrant vascularity. Lipid droplet (LD) accumulation is a hallmark of hypoxic cancer cells, yet how LDs form and function during hypoxia remains poorly understood. Herein, we report that in various cancer cells upon oxygen deprivation, HIF-1 activation down-modulates LD catabolism mediated by adipose triglyceride lipase (ATGL), the key enzyme for intracellular lipolysis. Proteomics and functional analyses identified hypoxia-inducible gene 2 (HIG2), a HIF-1 target, as a new inhibitor of ATGL. Knockout of HIG2 enhanced LD breakdown and fatty acid (FA) oxidation, leading to increased ROS production and apoptosis in hypoxic cancer cells as well as impaired growth of tumor xenografts. All of these effects were reversed by co-ablation of ATGL. Thus, by inhibiting ATGL, HIG2 acts downstream of HIF-1 to sequester FAs in LDs away from the mitochondrial pathways for oxidation and ROS generation, thereby sustaining cancer cell survival in hypoxia.
DOI: https://doi.org/10.7554/eLife.31132.001

*For correspondence:
liu.jun@mayo.edu

Competing interests: The authors declare that no competing interests exist.

## Introduction

Lipid droplets (LDs) are key organelles responsible for storing cellular surplus of fatty acids (FAs) in esterified forms such as triglycerides (TGs) and sterol esters (*Wilfling et al., 2014*). While TG-LDs likely form through de novo synthesis of TGs as a lens within the ER bilayer, TG catabolism/lipolysis is catalyzed by LD-localized adipose triglyceride lipase (ATGL) (*Zechner et al., 2012*). ATGL is the rate-limiting intracellular TG hydrolase in various cell and tissue types, and its activity in vivo is modulated by coactivator Comparative Gene Identification 58 (CGI-58) and inhibitor G0/G1 Switch Gene 2 (G0S2) (*Lass et al., 2006*; *Yang et al., 2010*). The enzymatic action of ATGL channels hydrolyzed FAs to mitochondria for β-oxidation as well as to the synthesis of lipid ligands for PPARα, whose activation in turn leads to enhanced mitochondrial biogenesis and function. In normal oxidative cell types such as hepatocytes, brown adipocytes and cardiomyocytes, loss of ATGL is known to cause accumulation of TG-LDs and impairment of mitochondrial oxidative capacity (*Ahmadian et al., 2011*; *Haemmerle et al., 2011*; *Ong et al., 2011*).

Recently, emerging evidence also points to a novel tumor suppressive role for ATGL. For example, whole-body ablation of ATGL in mice induces spontaneous pulmonary neoplasia (*Al-Zoughbi et al., 2016*). In addition, adipose-specific knockout of ATGL together with HSL causes

liposarcoma (*Wu et al., 2017*), and intestine-specific disruption of the ATGL co-activator CGI-58 promotes colorectal tumorigenesis (*Ou et al., 2014*). Collectively, these new findings implicate the possibility that inhibition of ATGL-mediated lipolysis may facilitate cancer development. However, the pathophysiological context and the molecular pathways that regulate ATGL in cancer are still unknown.

During the growth of solid tumors, hypoxic regions often arise when the rate of tumor cell proliferation exceeds that of angiogenesis (*Jain, 1988*). Hypoxia is a potent microenvironmental factor promoting aggressive malignancy, and is known for its association with poor survival in a variety of tumor types (*Hanahan and Weinberg, 2011*; *Masson and Ratcliffe, 2014*; *Rankin and Giaccia, 2016*). In response to oxygen deprivation, hypoxia inducible factors (HIFs) mediate multiple protective mechanisms, which together act to maintain oxygen homeostasis through reducing oxidative metabolism and oxygen consumption (*Gordan and Simon, 2007*; *Papandreou et al., 2006*; *Rankin and Giaccia, 2008*; *Semenza, 2010*).

To date, the best-characterized metabolic adaptation is the HIF-1-mediated switch from glucose oxidation to glycolysis for energy production (*Masson and Ratcliffe, 2014*; *Nakazawa et al., 2016*). In comparison, little is known regarding the regulation of intracellular fatty acid (FA) availability and oxidation in hypoxic cancer cells. Since enhanced fatty acid oxidation (FAO) and hypoxia both promote mitochondrial generation of reactive oxygen species (ROS) (*Bleier and Dröse, 2013*; *Guzy et al., 2005*; *Schönfeld and Wojtczak, 2008*), one conceivable mechanism for hypoxic cells to prevent oxidative stress is to channel free FAs to TG-LDs for storage. Indeed, increased accumulation of TG-LDs is now being recognized as a hallmark of hypoxic cancer cells of various origins (*Koizume and Miyagi, 2016*). Evidence derived from studies of cancer, glial and neural stem cells further implies that the capacity to accumulate LDs is positively linked to the ability of cells to survive oxidative stress in hypoxia (*Bailey et al., 2015*; *Bensaad et al., 2014*; *Liu et al., 2015*). However, establishing a definitive relationship would require a better understanding of the molecular mechanisms that govern hypoxia-induced LD formation.

In the present study, we have obtained compelling evidence to show that hypoxia causes profound inhibition of ATGL-mediated lipolysis in cancer cells. By using an unbiased proteomics screen and functional analyses, we identified a small protein encoded by Hypoxia-Inducible Gene 2 (HIG2) as a novel endogenous inhibitor of ATGL and as being solely responsible for mediating lipolytic inhibition in hypoxia. Our results further demonstrate that through inhibiting ATGL and mitochondrial FA oxidation, HIG2 acts downstream of HIF-1 to promote LD accumulation, attenuate ROS production, and enhance cancer cell survival in hypoxia.

## Results

### Intracellular lipolysis is reduced in hypoxic cancer cells

To determine whether lipolytic changes contribute to LD accumulation in hypoxia, we measured free FA release as an index of intracellular lipolysis in various human colorectal cancer (CRC) and renal cell carcinoma (RCC) cell lines under different oxygenated conditions. In ACHN (RCC), Caki-1 (RCC), DLD-1 (CRC) and HCT116 (CRC) cells, a 24 hr hypoxic treatment (0.5% $O_2$) resulted in release of significantly lower levels of FA (2.5–3-fold) relative to the normoxic condition (20% $O_2$) (*Figure 1A*). In response to hypoxia, the reduction in FA release was accompanied by an increased TG accumulation (2–3-fold) (*Figure 1B*). Genetic disruption of ATGL using CRISPR/Cas9 method, though profoundly decreased FA release in normoxic cells, failed to further reduce FA efflux in hypoxic HCT-116 cells (*Figure 1C*). In addition, hypoxia elicited no considerable changes in the expression of ATGL (*Figure 1D and G*) or its co-activator CGI-58 (*Lass et al., 2006*) (*Figure 1E and G*) and inhibitor G0S2 (*Yang et al., 2010*) (*Figure 1F*). These results indicate that ATGL-mediated lipolysis is suppressed in hypoxia *via* a mechanism independent of expressional regulation.

### HIG2 is identified as an ATGL interacting protein

To search for potential protein regulator(s) of ATGL, we expressed FLAG-ATGL in HCT116 cells and performed anti-FLAG immunoprecipitation after hypoxic treatment. Following resolution of co-immunoprecipitated proteins by SDS-PAGE (*Figure 2A*), Mass Spectrometry analysis identified one potential ATGL-binding partner as HIG2 (*Figure 2B and C*), a 63-amino acid (~7 kDa) protein

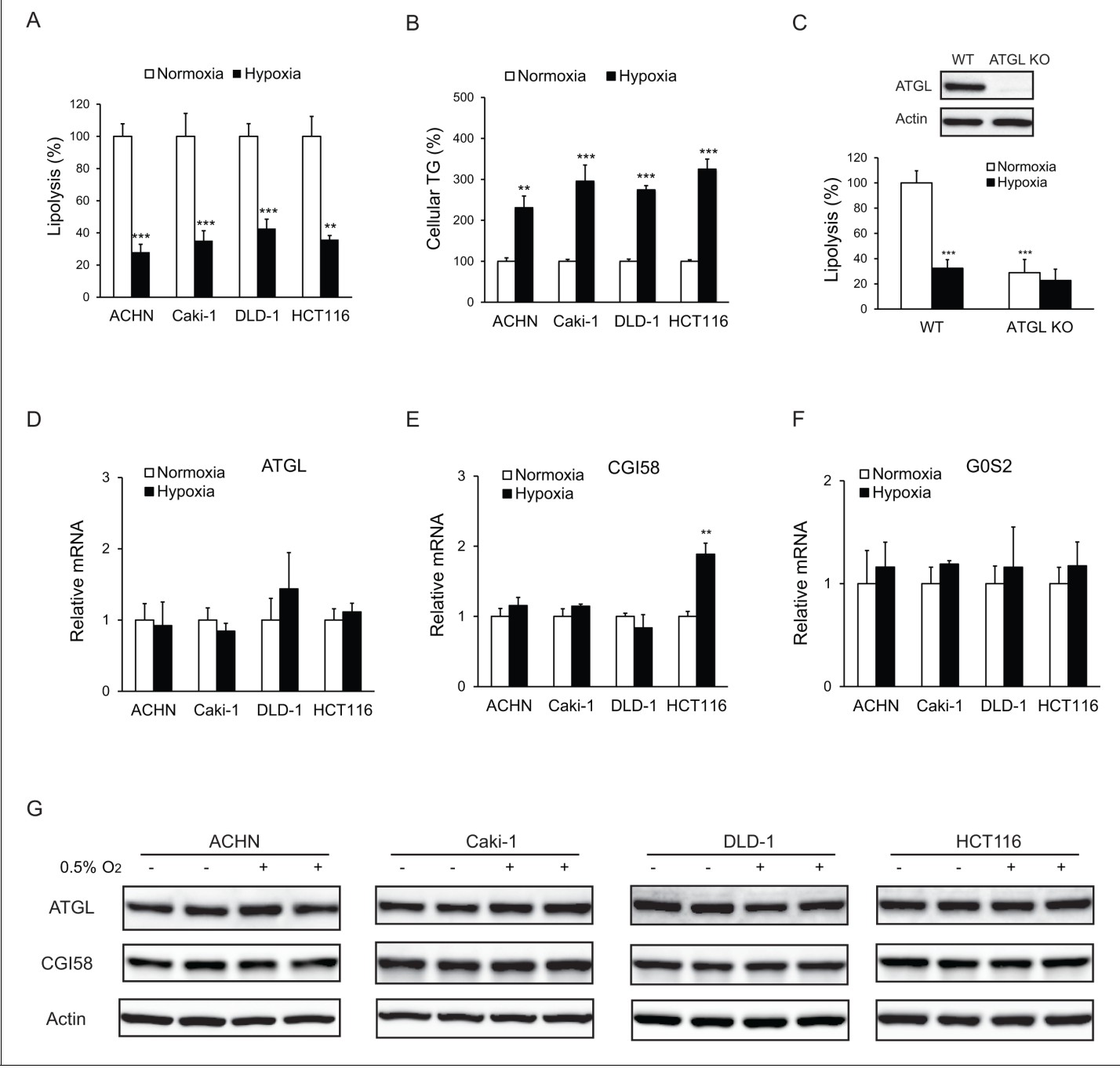

**Figure 1.** Lipolysis is reduced in cancer cells under hypoxia. (**A, B**) After 24 hr of incubation under normoxia or hypoxia, lipolysis (**A**) and cellular TG content (**B**) were measured. $n = 4$ biologically independent experiments. **$p<0.01$, ***$p<0.001$ vs. Normoxia. (**C**) Lipolysis in HCT116 clone cells after 24 hr of incubation under normoxia or hypoxia. ATGL knockout (KO) cells were generated by CRSPR/Cas9 method. $n = 3$ biologically independent experiments. ***$p<0.001$ vs. Normoxia WT. (**D–F**) mRNA levels of genes related to lipolysis in cells after 24 hr of incubation under normoxia or hypoxia. $n = 4$ biologically independent experiments. **$p<0.01$ vs. Normoxia. (**G**) Protein levels of ATGL and CGI58 in cells after 24 hr of incubation under normoxia or hypoxia were determined by immunoblotting with anti-ATGL and anti-CGI58 antibodies, respectively. Graphs represent mean ±SD, and were compared by two-tailed unpaired Student $t$-test.

DOI: https://doi.org/10.7554/eLife.31132.002

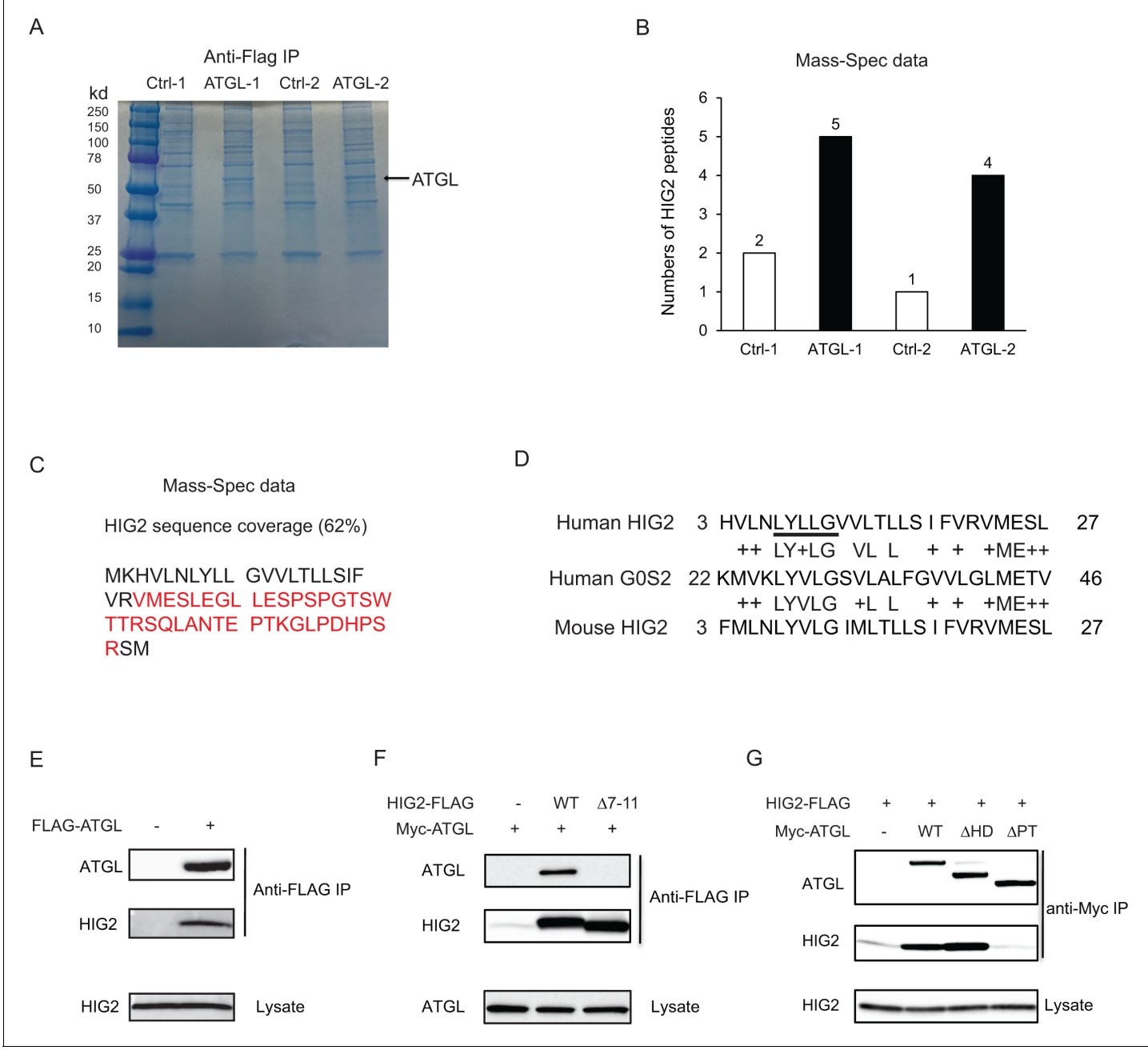

**Figure 2.** HIG2 interacts with ATGL. (**A**) HCT116 cells transfected with control vector or FLAG-ATGL vector were incubated under hypoxia for 24 hr followed by immunoprecipitation with anti-FLAG gels. Then the elution was separated on the 10–20% SDS-PAGE gel and stained with Coomassie Blue. The arrow indicates FLAG-ATGL. (**B**) The numbers of peptides from (**A**) recovered by mass spectrometry. (**C**) Combined coverage map of HIG2 peptides from (**B**). Detected peptides are in red. (**E**) Protein sequence alignment of HIG2 and G0S2. (**E**) HIG2 and ATGL in samples from (**A**) were detected by immunoblotting with anti-HIG2 and anti-FLAG antibodies, respectively. (**F**) HeLa cells were co-transfected with Myc-ATGL vector (N-terminal Myc tag) along with vector alone, HIG2-FLAG WT or HIG2-FLAG Δ7–11 vector (C-terminal FLAG tag). Immunoprecipitation was performed by anti-FLAG gels. HIG2 and ATGL proteins were detected by anti-FLAG and anti-Myc antibodies, respectively. (**G**) HeLa cells were co-transfected with HIG2-FLAG vector along with vector alone, Myc-ATGL or mutant vectors. ATGLΔPT and ATGLΔHD are two internal deletion mutants that lack the patatin domain (residues 10–178) and the hydrophobic domain (residues 259–337), respectively. Immunoprecipitation was performed by anti-Myc gels. HIG2 and ATGL proteins were detected by anti-FLAG and anti-Myc antibodies, respectively.

DOI: https://doi.org/10.7554/eLife.31132.003

encoded by the Hypoxia Inducible Lipid Droplet Associated (Hilpda) gene (*Gimm et al., 2010*). Interestingly, successive sequence alignment revealed that HIG2 contains a hydrophobic domain (HD) highly similar to the ATGL inhibitory domain of G0S2 (*Cerk et al., 2014*; *Yang et al., 2010*) (*Figure 2D*), raising the possibility of HIG2 being a novel ATGL inhibitor. Immunoblotting analysis provided the first piece of evidence that verified coimmunoprecipitation of ATGL with endogenous HIG2 (*Figure 2E*). The interaction between ATGL and HIG2 was further confirmed in HeLa cells (a human UCC cell line), when the two proteins were coexpressed (*Figure 2F*). Deletion of LY(V/L)LG (Δ7–11), a motif conserved between the HDs of HIG2 and G0S2, completely eliminated the ability of HIG2 to bind ATGL (*Figure 2F*). Moreover, ATGLΔPT and ATGLΔHD are two internal deletion mutants that lack the catalytic patatin-like domain (residues 10–178) and the LD-localizing hydrophobic domain (residues 259–337), respectively. As shown in *Figure 2G*, HIG2 co-immunoprecipitated with wild type ATGL and ATGLΔHD, while ATGLΔPT mutant exhibited no interaction with HIG2 (*Figure 2G*). Therefore, like G0S2 (*Yang et al., 2010*), HIG2 binds to the patatin-like catalytic domain of ATGL.

## HIG2 inhibits the TG hydrolase activity of ATGL

To determine if HIG2 regulates ATGL enzymatic activity, cell extracts of HeLa cells overexpressing ATGL were used as a source of ATGL in a TG hydrolase activity assay. As shown in *Figure 3A and B*, addition of in vitro translated HIG2 protein inhibited the activity of human and mouse ATGL by 80% and 85%, respectively. Similar extent of inhibition was observed when we included in the reaction either recombinant HIG2 (His-MBP-HIG2) purified from *E. coli* (*Figure 3C and D*) or HIG2-containing HeLa cell extracts (*Figure 3E*). HIG2 appears to be selective for ATGL, as it was unable to affect the TG hydrolase activity of hormone-sensitive lipase (HSL) (*Figure 3F*).

Immunofluorescence microscopy revealed that intracellular LD degradation mediated by ATGL was also effectively blocked by HIG2. As revealed by staining with BODIPY 493/503, a nonpolar probe selective for neural lipids such as TG, HeLa cells transfected with Myc-ATGL alone exhibited a marked reduction in both size and number of LDs upon oleic acid loading when compared with the adjacent untransfected cells (*Figure 3G*). However, co-expression of HIG2-FLAG was able to reverse this effect of Myc-ATGL. Consequently, HIG2-FLAG and Myc-ATGL were found to be co-localized at the surface of LDs. Furthermore, HIG2Δ7–11, a mutant deficient in interacting with ATGL, was incapable of preventing ATGL-induced LD degradation (*Figure 3G*), indicating the requirement of a direct interaction for ATGL inhibition.

## Lipolytic inhibition by HIG2 contributes to TG-LD accumulation and cancer cell survival under hypoxia

Early investigations found that overexpression of HIG2 promotes LD accumulation in various cell types (*DiStefano et al., 2015*; *Gimm et al., 2010*). Recently, a study using a conditional knockout mouse model showed that HIG2 mediates neutral lipid accumulation in macrophages and contributes to atherosclerosis in apolipoprotein E-deficient background (*Maier et al., 2017*). To determine whether inhibition of ATGL constitutes a major underlying mechanism, we used CRISPR/Cas9 method to disrupt HIG2 in HCT116, DLD-1, ACHN and HeLa cells. In cells from the control clones of all four cell lines, hypoxia dramatically induced HIG2 expression along with TG deposition (*Figure 4—figure supplement 1A–F*). While deletion of HIG2 mildly decreased the low levels of TG in normoxia, the ability to accumulate TG in hypoxia was uniformly lost in cells from the HIG2 knockout (KO) clones generated by using two independent gRNA (*Figure 4—figure supplement 1C–F*). Strikingly, co-disruption of ATGL was able to restore hypoxia-induced TG and LD accumulation in HIG2 KO cells (*Figure 4A–C*). Similar effect was achieved in the HIG2-deficient HCT116 cells by the overexpression of wild type HIG2 but not HIG2Δ7–11, the mutant disabled in ATGL interaction and inhibition (*Figure 4D*). Additionally, deletion of HIG2 alone in hypoxic cells led to a 7–fold increase in the lipolytic rate, which was completely reversed upon co-deletion of ATGL (*Figure 4E*). Taken together, these results provide proof that HIG2 acts to enhance intracellular TG levels through inhibiting ATGL-catalyzed lipolysis.

One of the most notable changes elicited by HIG2 deficiency was the decreased number of viable cells when hypoxia was prolonged. While the growth of the wild type cells was generally decreased in hypoxia, disruption of HIG2 or/and ATGL incurred no further changes within a 24 hr period of

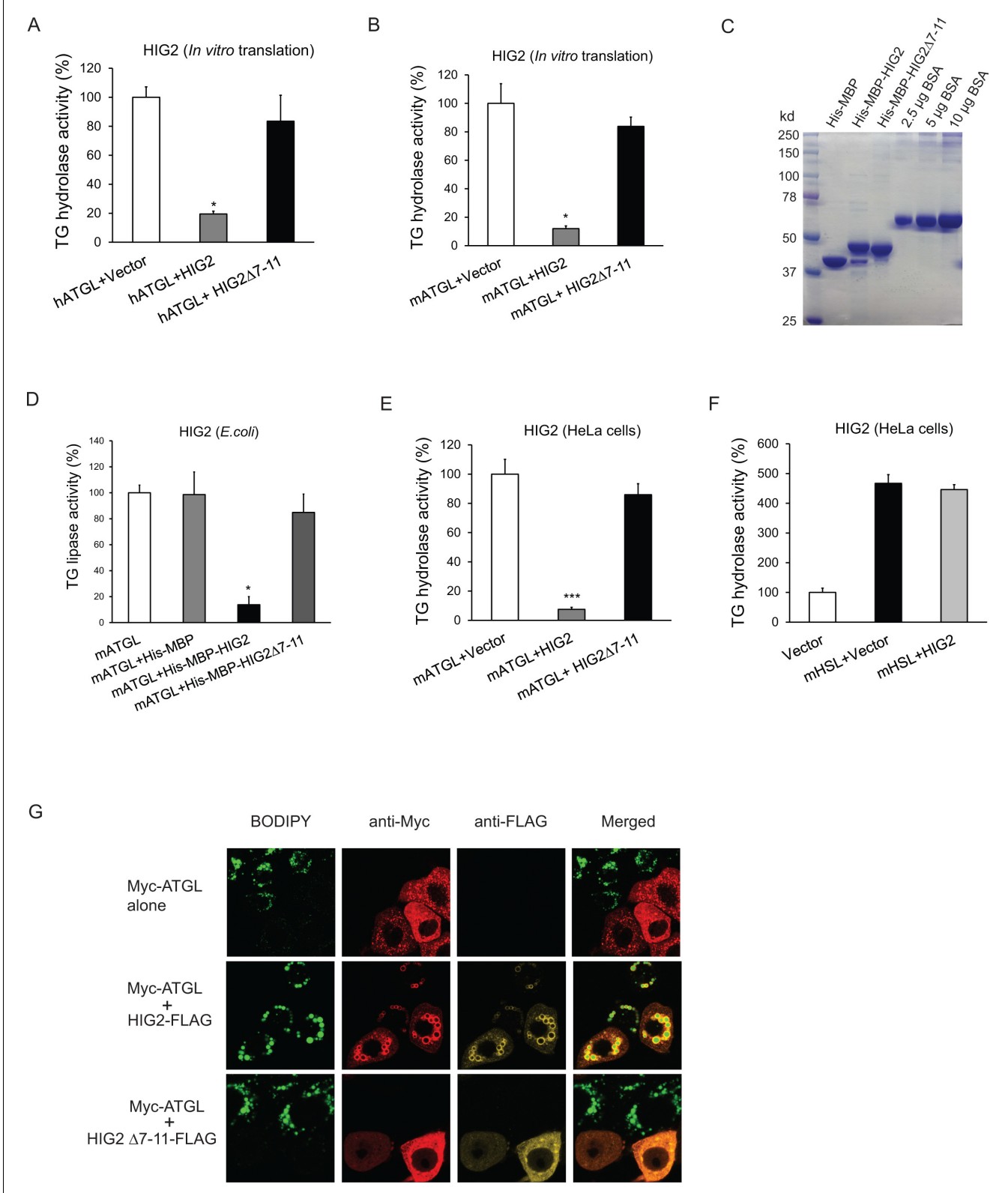

**Figure 3.** HIG2 inhibits ATGL enzymatic activity. (**A, B**) HIG2 from in vitro translation was added to extracts of HeLa cells transfected with human ATGL vector (hATGL) (**A**) or mouse ATGL vector (mATGL) (**B**), and TG hydrolase activity was determined. *n* = 2 biologically independent experiments. *p<0.05 vs. hATGL +Vector or mATGL +Vector. (**C**) 10 µl of recombinant proteins purified from E. coli were separated on a 10% SDS-PAGE gel and stained with Coomassie Blue. (**D**) 4 µg of recombinant proteins from (**c**) were added to extracts of HeLa cells transfected with mouse ATGL vector, and TG hydrolase

*Figure 3 continued on next page*

Figure 3 continued

activity was determined. *n* = 4 biologically independent experiments. *n* = 2 biologically independent experiments. *p<0.05 vs. mATGL +His MBP. (**E, F**) Lysate from HeLa cells transfected with HIG2 vectors was added to extracts of HeLa cells expressing mATGL (**E**) or mHSL (**F**), and TG hydrolase activity was measured. *n* = 4 biologically independent experiments. ***p<0.001 vs. mATGL +Vector. (**G**) HeLa cells transfected with Myc-ATGL vector in the absence or presence of HIG2-FLAG WT or HIG2-FLAG Δ7–11 vector were incubated with 200 µM of oleate complexed to BSA overnight followed by immunofluorescence staining. Lipid droplets were stained by BODIPY 493/503 (green). Graphs represent mean ±SD, and were compared by two-tailed unpaired Student *t*-test.

DOI: https://doi.org/10.7554/eLife.31132.004

hypoxic treatment (*Figure 5A and B*). However, when hypoxia was extended to 48 hr, loss of HIG2 caused a significant reduction in the number of viable cells (*Figure 5B*) as well as a marked increase in apoptotic cell death, as evidenced by the robust appearance of cleaved PARP and Caspase-3 proteins (*Figure 5C*; *Figure 5—figure supplement 1A–E*). Fluorescence-activated cell sorting (FACS) analysis further demonstrated that HIG2 disruption increased the rate of apoptosis from 3.73% to 14.6% after extended hypoxia (*Figure 5D*; *Figure 5—figure supplement 1F*). In HIG2 KO cells, deletion of ATGL increased the number of viable cells comparable to that of wild type cells (*Figure 5B*). Loss of ATGL also completely rescinded the cleavage of PARP and Caspase-3 (*Figure 5C*; *Figure 5—figure supplement 1E*) as well as substantially reduced the number of apoptotic cells (*Figure 5D*; *Figure 5—figure supplement 1F*). In addition, apoptosis induced by HIG2 deficiency could be recued by the overexpression of wild type HIG2 but not HIG2Δ7–11 as revealed by immunoblotting (*Figure 5E*) and FACS analysis (*Figure 5F*). Therefore, through inhibiting ATGL, HIG2 plays an essential role in protecting against apoptosis and thus sustaining cell survival during extended hypoxia.

## Lipolytic inhibition promotes hypoxic cell survival through reducing PPARα activity, FAO and ROS production

ATGL is known to be a key regulator of PPARα activation and mitochondrial FA oxidation (FAO) in normal oxidative cell types (*Zechner et al., 2012*). In normoxic HCT116 cells that express low levels of HIG2 protein, deletion of ATGL or/and HIG2 caused no significant differences in the mRNA levels of *Ppara* and its target genes for FAO including *Cpt1a*, *Cpt1b*, *Vlcad*, *Acaa2* and *Mcad* (*Figure 6A*) or the rates of FAO as measured by the rate of the production of radiolabeled $H_2O$ from radiolabeled oleic acid (*Figure 6B*). In response to hypoxia, the wild type and ATGL KO cells displayed a pronounced decrease in both the rates of FAO and the expression of PPARα and its target genes (*Figure 6A and B*). By contrast, hypoxic HIG2 KO cells largely maintained the expression of FAO genes and levels of FAO. These effects were consistent regardless of whether radiolabeled oleic acid was added to the cells during hypoxia or intracellular TG was pre-labeled in normoxia prior to the cells being exposed to hypoxia (*Figure 6—figure supplement 3A*). Co-deletion of ATGL was able to rescue these effects of HIG2 deficiency (*Figure 6A and B*), arguing that HIG2-mediated ATGL inhibition, instead of the decreased oxygen supply, is primarily responsible for the diminished FAO in hypoxia. Interestingly, loss of HIG2 does not appear to affect glycolytic phenotypes as hypoxia induced similar increases of glucose consumption and lactate production in wild type and HIG2 KO cells (*Figure 6—figure supplement 1A–D*). Thus, the inhibition of FA mobilization by HIG2 does not appear to impact glycolytic phenotypes in hypoxic cancer cells.

Enhanced FAO and hypoxia both promote mitochondrial generation of ROS (*Bleier and Dröse, 2013*; *Guzy et al., 2005*; *Schönfeld and Wojtczak, 2008*). We speculated that in hypoxic HIG2 KO cells, increased FAO and low oxygen conditions would synergistically cause excessive ROS production. In support of this hypothesis, HIG2 KO cells exhibited a near 250% increase of intracellular ROS levels in hypoxia as compared to normoxia (*Figure 6C*; *Figure 6—figure supplement 2*). This is in contrast to the wild type, ATGL KO and HIG2/ATGL double knockout (dKO) cells, all of which only experienced a ~120% increase in ROS production in hypoxia (*Figure 6C*; *Figure 6—figure supplement 2*). Most importantly, treatment of hypoxic HIG2 KO cells with anti-oxidant N-acetyl cysteine (NAC) dose-dependently inhibited cleavage of PARP and Caspase-3 (*Figure 6D*) and cell labeling by Annexin V (*Figure 6E*). Interestingly, the PPARα antagonist, GW6471, also inhibited both ROS production and apoptosis (*Figure 6F and G*). Treatment of cells with Ranolazine or Trimetazidine, two different pharmaceutical inhibitors of FAO, similarly blocked apoptosis in HIG2 KO cells (*Figure 6H–*

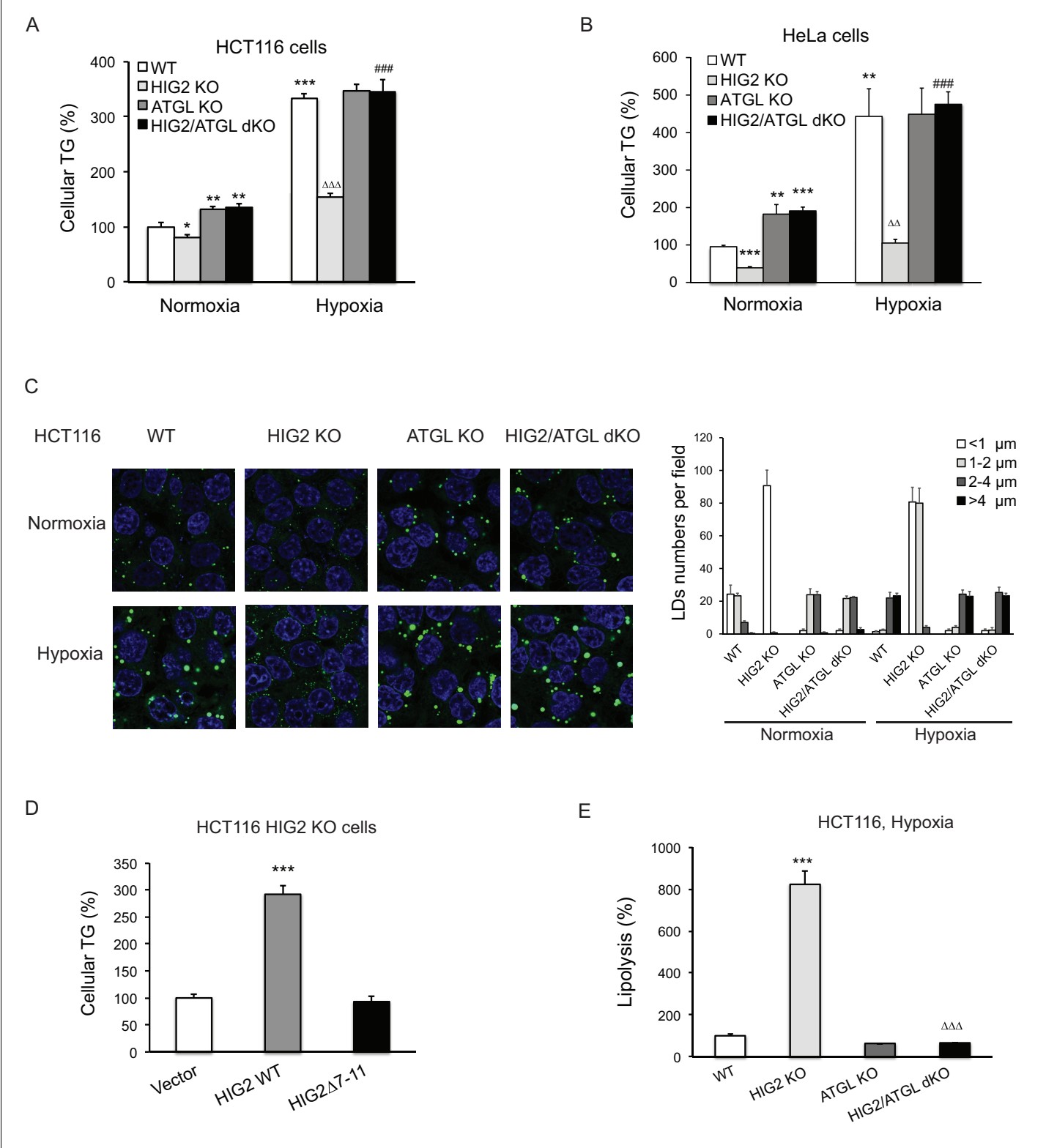

**Figure 4.** Lipolytic inhibition by HIG2 causes TG accumulation under hypoxia. (**A–C**) HCT116 or HeLa KO clone cells created by CRISPR/Cas9 method were incubated under normoxia or hypoxia for 24 hr, and then cellular TG was detected by TG kits or by BODIPY 493/503 (green indicates lipids and blue indicates DAPI-stained nucleus). *n* = 3 biologically independent experiments for (**A**); *n* = 4 biologically independent experiments for (**B**).*p<0.05, **p<0.01, ***p<0.001 vs. Normoxia WT; ^ΔΔ^p<0.01, ^ΔΔΔ^p<0.001 vs. Hypoxia WT; ^###^p<0.001 vs. Hypoxia HIG2 KO. (**D**) Following an overnight transfection with DNA vectors, HCT116 cells were incubated under normoxia or hypoxia for 24 hr, and then TG was measured. *n* = 4 biologically independent

*Figure 4 continued on next page*

*Figure 4 continued*

experiments. \*\*\*p<0.001 vs. Vector. (E) Lipolysis in HCT116 clone cells after 24 hr of incubation under hypoxia. *n* = 4 biologically independent experiments. \*\*\*p<0.001 vs. WT; ᐃᐃᐃp<0.001 vs. HIG2 KO. Graphs represent mean ±SD, and were compared by two-tailed unpaired Student *t*-test.
DOI: https://doi.org/10.7554/eLife.31132.005
The following figure supplement is available for figure 4:

**Figure supplement 1.** Enhancement of lipolysis reduces TG accumulation under hypoxia.
DOI: https://doi.org/10.7554/eLife.31132.006

*6J*; *Figure 6—figure supplement 3B*). Therefore, in the absence of HIG2, PPARα activation of FAO downstream of ATGL action may cause elevation of ROS levels and promote apoptotic induction by hypoxia.

## Effects of HIF-1 on lipid metabolism and cell survival are dependent on lipolytic inhibition

Next, we determined whether HIF-1 acts upstream of HIG2 in initiating the pathway that leads to the inhibition of ATGL-mediated lipolysis and FAO. In line with *HIG2* as a target gene of HIF-1, knockdown of HIF-1α using a specific siRNA oligo caused a substantial decrease in HIG2 expression induced by hypoxia (*Figure 6—figure supplement 4A*). Reminiscent of HIG2 ablation, HIF1α knockdown restored lipolysis, decreased TG accumulation and enhanced FAO in the wild type cells under hypoxic conditions (*Figure 6—figure supplement 4B–D*). By contrast, these effects incurred by HIF-1 knockdown were absent in the ATGL KO cells. In response to hypoxia, intracellular ROS levels (*Figure 6—figure supplement 4E*) and cell apoptosis (*Figure 6—figure supplement 4A* and *Figure 6—figure supplement 4F*) were also markedly increased by HIF-1α knockdown in the wild type but not ATGL KO cells, though both cell types exhibited reduced HIG2 expression upon knockdown of HIF-1α. Collectively, these findings establish the previously uncharacterized antilipolytic role of a HIF-1α-HIG2 axis in the protection of hypoxic cells from ROS-induced cell death.

## Lipolytic inhibition is critical for tumor growth in vivo.

To determine the in vivo role of lipolytic inhibition mediated by HIG2, we injected wild type, ATGL KO, HIG2 KO, and HIG2/ATGL dKO HCT116 cells subcutaneously into nude mice to generate tumor xenografts. Deletion of HIG2 resulted in a profound delay in tumor growth as compared to the wild type control group (*Figure 7A*). In particular, we observed that tumors in the wild type group reached volumes of ~1100 mm$^3$ (>600 mg in weight) after only 25 days, whereas tumor volumes in the HIG2 KO group were only ~180 mm3 (<100 mg in weight) (*Figure 7B and C*). Histological analysis of tissue sections revealed a substantially reduced accumulation of neutral lipids, increased cleavage of Caspase-3 and increased staining of the lipid peroxidation marker 4-HNE in the HIG2 KO tumors (*Figure 7D*; *Figure 7—figure supplement 1*). In contrast, loss of ATGL alone elicited no significant changes in either tumor growth or intra-tumor lipid accumulation (*Figure 7A–D*; *Figure 7—figure supplement 1*). Interestingly, disruption of ATGL along with HIG2 restored tumor growth and rescued the effects elicited by HIG2 disruption alone (*Figure 7A–D*; *Figure 7—figure supplement 1*). Similar results were obtained when cells from different HeLa cell clones were used to produce tumor xenografts (*Figure 7E and F*). These results demonstrate that inhibition of lipolysis by HIG2 promotes tumor lipid accumulation, thereby preventing oxidative stress-induced apoptosis and promoting tumor survival in vivo.

## The lipolytic pathway is downregulated in human cancers

To determine if the lipolytic pathway is affected in human tumors, we analyzed solid tumor data sets in the TCGA (The Cancer Genome Atlas). Consistent with our RCC and CRC cell lines, we found that HIG2 mRNA abundance is strongly associated with various solid tumors including kidney renal clear cell carcinoma (KIRC), colon/rectum adenocarcinoma (COAD/READ), lung squamous cell carcinoma (LUSC), bladder urothelial carcinoma (BLCA), and uterine corpus endometrial carcinoma (UCEC), etc. (*Figure 8—figure supplement 1*). In COAD and KIRC, the expression pattern of HIG2 mirrors that of other signature target genes of HIF-1 such as LDHA, GLUT1 and VEGFA (i.e. dormant in normal tissues but highly upregulated in tumors)(*Figure 8A*). To confirm increased expression of HIG2

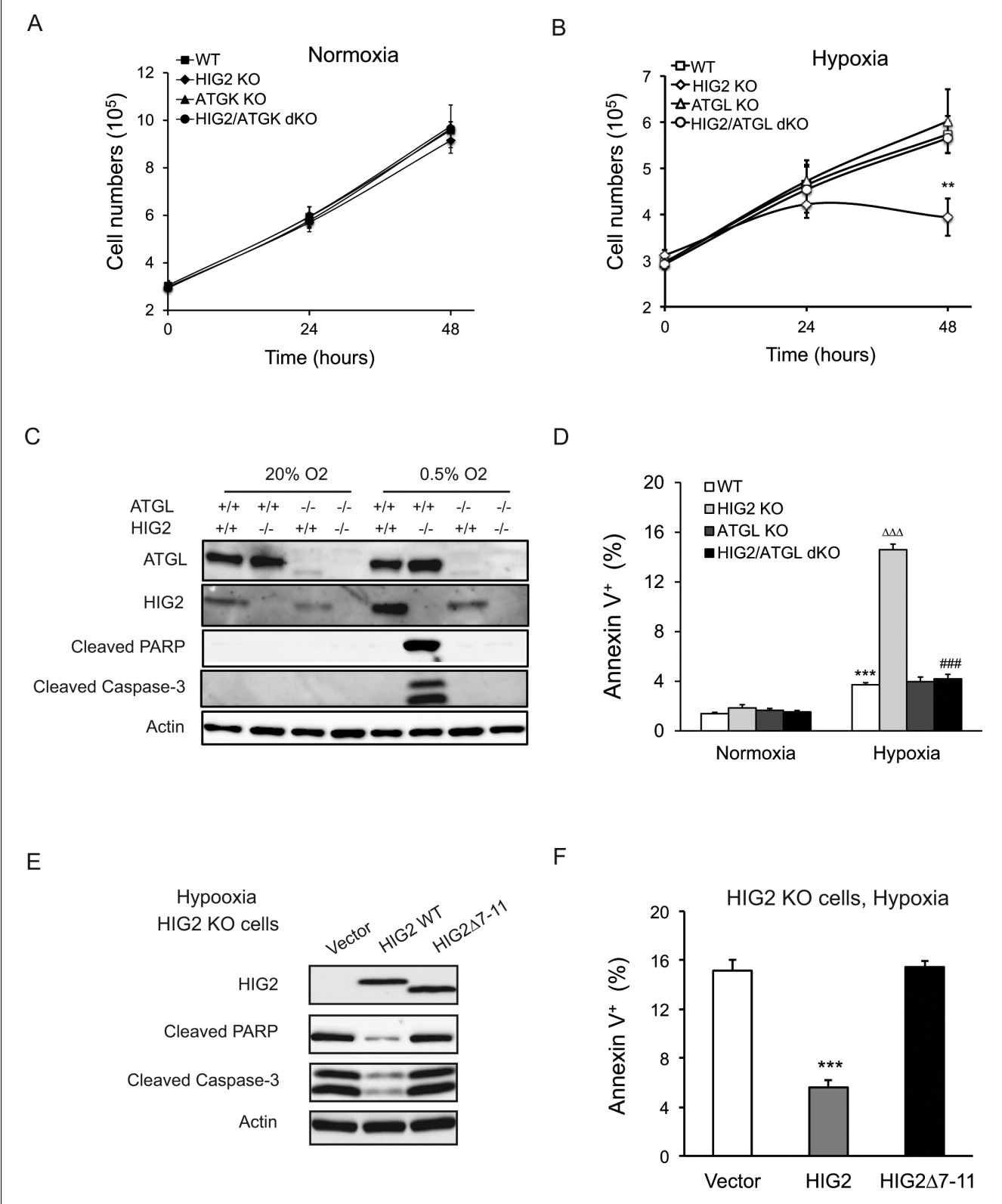

**Figure 5.** Lipolytic inhibition by HIG2 prevents cell apoptosis under hypoxia. (**A, B**) Cell number was determined by counting viable cells at indicated times. $n$ = 4 biologically independent experiments. **p<0.01 vs. Hypoxia WT. (**C, D**) After 48 hr of incubation under normoxia or hypoxia, apoptosis in HCT116 clone cells was assessed by immunoblotting (**C**) or by staining with Annexin V for Flow Cytometry (**D**). $n$ = 3 biologically independent experiments. ***p<0.001 vs. Normoxia WT; ΔΔΔp<0.001 vs. Hypoxia WT; ###p<0.001 vs. Hypoxia HIG2 KO. (**E, F**) Following an overnight transfection with

*Figure 5 continued on next page*

*Figure 5 continued*

DNA vectors, HCT116 cells were cultured under normoxia or hypoxia for 48 hr. Cell apoptosis was examined under hypoxia by immunoblotting (**E**) or by Flow Cytometry (**F**). *n* = 2 biologically independent experiments. ***p<0.001 vs. Vector. Graphs represent mean ±SD, and were compared by two-tailed unpaired Student *t*-test.

DOI: https://doi.org/10.7554/eLife.31132.007

The following figure supplement is available for figure 5:

**Figure supplement 1.** Enhancement of lipolysis induces apoptosis under hypoxia.

DOI: https://doi.org/10.7554/eLife.31132.008

protein in vivo, we compared levels of HIG2 in 19 RCC samples matched by Fuhrman grade (grade 2) and by adjacent uninvolved kidney tissue. HIG2 protein was highly expressed in RCC tissues but hardly detectable in the matched adjacent normal kidney tissue, coinciding with the HIF-1α protein levels and tissue TG content (*Figure 8B*; *Figure 8—figure supplement 2*). By contrast, no significant differences were detected in the protein expression of ATGL or CGI-58 (*Figure 8B*). These observations indicate that the lipolytic inhibition mediated by the upregulation of HIG2 is a relevant mechanism for cancer pathophysiology in humans.

## Discussion

Understanding how cancer cells become adapted to hypoxia is central to understanding how hypoxia promotes tumor progression and malignancy. One compelling idea is that metabolic adaptations driven by HIF-1 confer a selective advantage for cancer cells in the low oxygen environment. It had been recognized previously that through enhanced HIF-1 activity, cancer cells increase their TG-LD accumulation in response to oxygen deprivation (*Bensaad et al., 2014*; *Koizume and Miyagi, 2016*). Additionally, HIF-1 activation downregulates mitochondrial oxidative capacity (*Masson and Ratcliffe, 2014*; *Zhang et al., 2007*), by which it helps to reduce oxygen consumption and maintain oxygen homeostasis in hypoxia. However, the earlier studies did not precisely define how HIF-1 functions to facilitate these two metabolic alterations. In this regard, the present study has uncovered a major unifying mechanism by demonstrating that inhibition of ATGL-mediated lipolysis by HIG2 contributes to LD storage and attenuated mitochondrial FA oxidation under hypoxia.

The first hint of the HIG2 protein function came from the sequence alignment that revealed a homology between the HIG2 HD and the ATGL inhibitory domain of G0S2. Biochemical and cell biological analyses subsequently confirmed that like G0S2, HIG2 specifically inhibits the TG hydrolase activity of ATGL. Decreased TG hydrolysis results from a specific interaction, since deletion of the LY (V/L)LG motif conserved between the HDs of HIG2 and G0S2 abolished both ATGL interaction and inhibition. Recently, Cerk et al. demonstrated that a peptide derived from the G0S2 HD containing the LY(V/L)LG motif is capable of inhibiting ATGL in a dose dependent, non-competitive manner (*Cerk et al., 2014*). It is highly conceivable that HIG2 inhibits ATGL *via* a similar biochemical mechanism. In addition, HIG2-ablated hepatocytes were previously shown to exhibit increased TG turnover under normoxic conditions (*DiStefano et al., 2015*). However, knockout mouse studies conducted by the same group and Dijk et al. later yielded results arguing against a direct involvement of HIG2 in lipolysis (*Dijk et al., 2017*; *DiStefano et al., 2016*). The reason for these discrepancies is currently unknown. We speculate that the lack of hypoxia in the employed experimental settings, under which endogenous HIG2 might be expressed at low levels and thus possess a relatively insubstantial role, may contribute to the absence of significant changes caused by HIG2 ablation.

Knockout of HIG2 increased lipolysis and decreased TG levels in hypoxic cancer cells. Co-ablation of ATGL was able to rescue these phenotypes of HIG2 deficiency, suggesting that the specific inhibition of ATGL is functionally important for the adaptive LD accumulation downstream of HIG2 expression. Moreover, early studies have shown that ATGL-mediated TG hydrolysis provides necessary lipid ligands for PPARα activation. Associated with a markedly diminished PPARα target gene expression, ATGL-deficient cells often exhibit severely disrupted mitochondrial oxidation of FAs (*Ahmadian et al., 2011*; *Haemmerle et al., 2011*; *Ong et al., 2011*). In agreement with these previous findings, our data demonstrate that in cancer cells, hypoxia-induced downregulation of FAO and reduction in expression of PPARα target genes both are dependent on HIG2. This dependency was completely lost upon ATGL ablation, again suggesting a prerequisite role for ATGL inhibition by

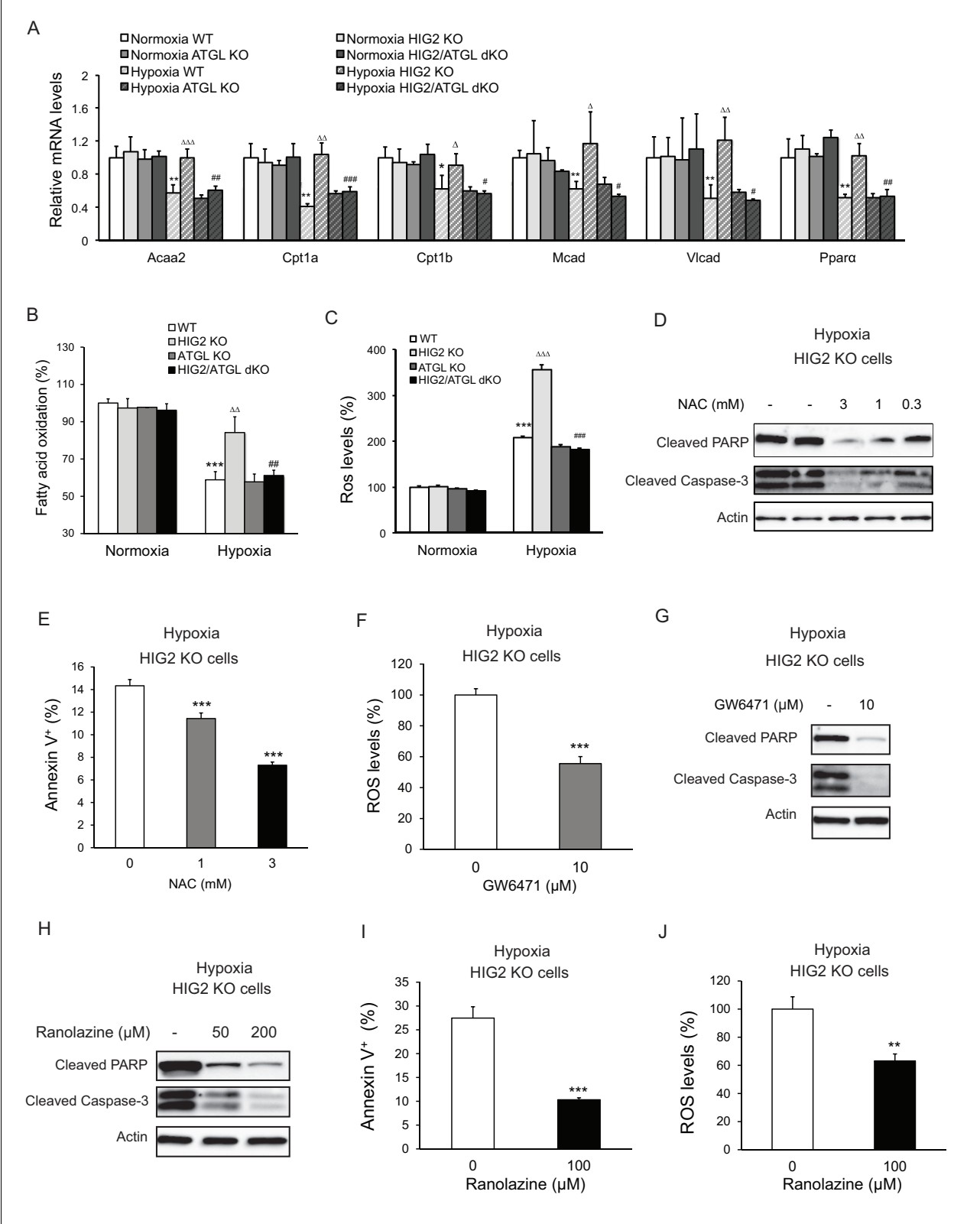

**Figure 6.** Enhancement of lipolysis in the absence of HIG2 increases PPARα activity, FAO rate and ROS production under hypoxia. (A–C) After 36 hr of incubation under normoxia or hypoxia, mRNA levels (A), FAO (B) and ROS levels (C) were measured in HCT116 clone cells. *Acaa2*, acetyl-CoA acyltransferase 2; *Cpt1a*, carnitine palmitoyltransferase Ia; *Cpt1b*, carnitine palmitoyltransferase Ib; *Mcad*, Medium-chain acyl-CoA dehydrogenase; *Vlcad*, very-long-chain acyl-CoA dehydrogenase. $n$ = 4 (WT, HIG KO) or 3 (ATGL KO, HIG2/ATGL KO) biologically independent experiments for (A);
*Figure 6 continued on next page*

*Figure 6 continued*

$n$ = 5 (WT, HIG KO) or 3 (ATGL KO, HIG2/ATGL KO) biologically independent experiments for (B); $n$ = 2 biologically independent experiments for (C). *p<0.05, **p<0.01, ***p<0.001 vs. Normoxia WT; $^\Delta$p<0.05, $^{\Delta\Delta}$p<0.01, $^{\Delta\Delta\Delta}$p<0.001 vs. Hypoxia WT; #p<0.05, ##p<0.01, ###p<0.001 vs. Hypoxia HIG2 KO. (D, E) HCT116 cells were cultured under hypoxia in the presence or absence of various concentrations of NAC for 48 hr. Cell apoptosis was determined by immunoblotting (D) or by staining with Annexin V for Flow Cytometry (e). $n$ = 3 biologically independent experiments. ***p<0.001 vs. without NAC treatment. (F) ROS levels in HCT116 cells were determined by $H_2DCFDA$ after 36 hr of treatment with 10 µM GW6471 under hypoxia. $n$ = 3 biologically independent experiments. ***p<0.001 vs. without GW6471 treatment. (G) Protein levels of apoptosis markers in HCT116 cells after 48 hr of treatment with 10 µM GW6471 under hypoxia. (H–J) HCT116 cells were cultured under hypoxia in the presence or absence of ranolazine. After 48 hr of treatment, cell apoptosis was determined by immunoblotting (H) or by staining with Annexin V for Flow Cytometry (I). Intracellular ROS levels (36 hr of treatment) were determined by $H_2DCFDA$ (J). $n$ = 3 biologically independent experiments. **p<0.01, ***p<0.001 vs. without ranolazine treatment. Graphs represent mean ±SD, and were compared by two-tailed unpaired Student $t$-test.

DOI: https://doi.org/10.7554/eLife.31132.009

The following figure supplements are available for figure 6:

**Figure supplement 1.** The effects of lipolysis on glucose metabolism.
DOI: https://doi.org/10.7554/eLife.31132.010

**Figure supplement 2.** ROS levels in cells of different HCT116 CRISPR clones.
DOI: https://doi.org/10.7554/eLife.31132.011

**Figure supplement 3.** Fatty oxidation and apoptosis in HCT116 cells.
DOI: https://doi.org/10.7554/eLife.31132.012

**Figure supplement 4.** Apoptosis induced by HIF-1α knockdown is lipolysis-dependent.
DOI: https://doi.org/10.7554/eLife.31132.013

HIG2. Inhibition of FAO and antagonism of PPARα both led to reduced ROS production and apoptotic induction in hypoxic HIG2 KO cells, indicating the activation of PPARα-dependent FAO as a major contributor to oxidative stress. PPARα is generally thought to limit lipotoxicity through upregulating FAO during excess FA availability. While sustained activation of FAO often results in increased generation of ROS in mitochondria, PPARα is known to promote the expression of various anti-oxidases such as catalase and superoxide dismutase (SOD) (*Khoo et al., 2013*; *Liu et al., 2012*). We speculate that in normoxia, mechanisms balancing ROS generation and degradation likely operate to maintain the steady-state redox environment. However, during hypoxia when oxygen is insufficiently supplied, a tilt toward excessive ROS production can occur as a result of increased electron leakage from the mitochondrial electron transport chain (ETC), leading to oxidative stress.

A question arises as to why lipolytic inhibition in hypoxic cancer cells would be advantageous. Cancer cells require large amounts of lipids for the synthesis of cellular membranes to maintain high cell proliferation rates. However, lipotoxicity can occur at times when FA delivery to the cells exceeds FA oxidation rates. This may be especially important during the periods of hypoxia, when HIF-1 activation is known to induce FA uptake (*Bensaad et al., 2014*). On the other hand, the oxidation of FAs may need to be decelerated in hypoxia as it consumes significant amounts of oxygen, which can exacerbate oxygen insufficiency. More importantly, both hypoxia and excessive FA oxidation cause elevated mitochondrial ROS production (*Bleier and Dröse, 2013*; *Guzy et al., 2005*; *Schönfeld and Wojtczak, 2008*). Elevated ROS levels lead to peroxidation of membrane lipids, denaturation of proteins and deactivation of enzymes, which together can lead to cell damage and apoptosis. Therefore, switch-off of FA oxidation combined with storage of excess FAs in TG-LDs through inhibition of lipolysis would constitute a conceivable strategy for cancer cells to prevent ROS overproduction and oxidative damage as well as evade lipotoxicity in hypoxia. In this regard, the present study provides compelling evidence that the HIF-1-HIG2 antilipolytic pathway is a central component of such a survival strategy. In hypoxia, restoration of lipolysis by ablation of either HIG2 or HIF-1α caused significant increases in FAO and ROS generation along with decreased cell viability. A causal relationship between ROS elevation and increased cell death is demonstrated by the fact that exogenously applied antioxidant NAC protected HIG2 KO cells from hypoxia-induced apoptosis. Furthermore, we found that overexpression of wild type HIG2 but not HIG2Δ7–11, the mutant deficient in ATGL inhibition, decreased ROS production and increased resistance to hypoxia-induced apoptosis. These results clearly establish that inhibition of ATGL-mediated lipolysis by HIG2 is downstream of and required for HIF-1 to elicit the protective effects in hypoxic cancer cells.

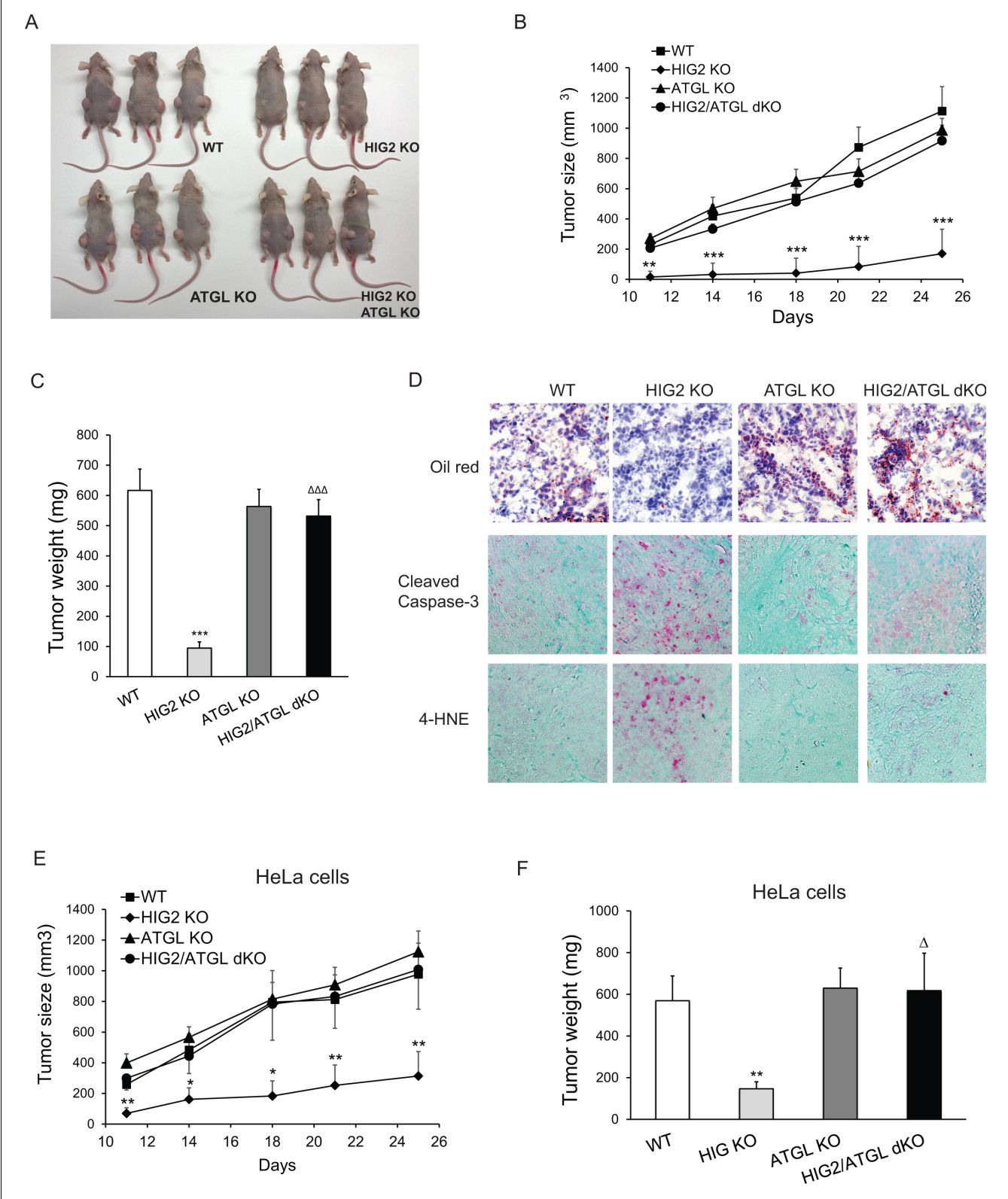

**Figure 7.** Enhancement of lipolysis delays tumor growth in vivo. (A–D) HCT116 clone cells were injected subcutaneously into both flanks of nude mice. (A) Imaging of representative mice at the end of the experiment. (B) Tumor growth curves were measured starting from 11 days after inoculation. (C) Tumors were extracted and weighted at the end of the experiment. (D) Representative imaging of oil red O staining, cleaved caspase-3 immunostaining and 4-HNE immunostaining. Data represent mean ±SEM from independent xenografts (WT, $n$ = 12; HIG KO, $n$ = 16; ATGL KO, $n$ = 10; *Figure 7 continued on next page*

*Figure 7 continued*

HIG2/ATGL KO, n = 12). **p<0.01, ***p<0.001 vs. WT; ^△△△p<0.001 vs. HIG2 KO. (E and F) HeLa clone cells were injected subcutaneously into both flanks of nude mice. (E) Tumor growth curves were measured starting from 11 days after inoculation. (F) Tumors were weighted at the end of the experiment. Data represent mean ±SEM from independent xenografts (WT, n = 12; HIG KO, n = 10; ATGL KO, n = 12; HIG2/ATGL KO, n = 8). *p<0.05, **p<0.01 vs. WT; ^△p<0.05 vs. HIG2 KO.
DOI: https://doi.org/10.7554/eLife.31132.014
The following figure supplement is available for figure 7:

**Figure supplement 1.** Quantification of oil red O staining, cleaved caspase-3 immunostaining and 4-HNE immunostaining in tumor xenografts.
DOI: https://doi.org/10.7554/eLife.31132.015

Our results complement the existing model illustrating that HIF-1 represses glucose flux to the TCA cycle through mediating expression of PDK1, which inhibits pyruvate oxidation through phosphorylating and inactivating PDH (*Kim et al., 2006*). In cancer cells located within the poorly oxygenated regions of solid tumors, coordinate inhibition of ATGL and PDH by HIG2 and PDK1, respectively, should collectively lead to reduced mitochondrial oxidative metabolism and ROS production as well as improved tissue oxygen homeostasis (*Figure 8C*). In addition, we observed that neither hypoxia-induced glucose consumption nor glycolysis was affected by HIG2 deletion. Although excessive mitochondrial FAO may impair glucose utilization *via* the classic Randle cycle, our results suggest that the acquisition of glycolytic phenotypes by hypoxic cancer cells is independent of the inhibition of FA mobilization by HIG2. We speculate that the main purpose of storing excessive FAs in TG-LDs in hypoxia likely is for prevention of the potential cytotoxicity that is associated with FAO-driven ROS generation.

Based on the data derived from the present study, we propose that HIG2 is a novel metabolic oncogenic factor, which exerts its function by neutralizing the tumor suppressive role of ATGL/CGI-58. Our observations that ATGL inhibition by HIG2 promotes hypoxic cancer cell survival in vitro and tumor growth in vivo are supportive of this hypothesis. By suggesting a critical role of lipolytic inhibition in hypoxic tumor areas, the present study provides justification for the development of specific chemical disruptors of HIG2-ATGL interaction. Such drugs presumably would be able to liberate ATGL and potentiate FAO-driven ROS production to toxic levels, resulting in apoptotic death of hypoxic cancer cells. Our findings that HIG2 is highly upregulated in multiple human solid tumors are in agreement of this concept. The HIF-1-HIG2 antipolytic pathway represents a departure from the typical metabolic pathways that have been targeted therapeutically to deprive cells of necessary fuel/building blocks. Lastly, it is important to note that while the bioinformatics analysis revealed an upregulated expression of HIG2 in a variety of solid tumors, our mechanistic studies mainly employed HeLa, CRC and RCC cell lines. It is conceivable that the impact of HIG2 as a lipolytic inhibitor varies among different tumor types. In this regard, increased expression of LD coat protein Perilipin 2 downstream of HIF-2 was recently shown to promote lipid storage in clear cell RCC (*Qiu et al., 2015*). Together, the accumulating data have painted a complex and intricate picture in which cancer cells regulate their lipid accumulation in hypoxia.

## Materials and methods

### Reagents

The following antibodies were used: Rabbit anti-HIG2 (Santa Cruz, #sc-137518, 1:500 dilution); Rabbit anti-HIF-1α (Novusbio, #NB100-449, 1:500 dilution); Rabbit anti-c-Myc (Santa Cruz, #sc-789, 1:1000 dilution); Mouse anti-Flag M2 (Sigma, #F1804, 1:1000 dilution); Mouse anti-Actin (Sigma, #A1978, clone AC-15, 1:10,000 dilution); Rabbit anti-cleaved Caspase-3 (Cell signaling technology, #9661, 1:300 dilution); Rabbit anti-cleaved PARP (Cell signaling technology, #5625, clone D64E10, 1:1000 dilution); Rabbit anti-ATGL (Cell signaling technology, #2138, 1:500 dilution); Rabbit anti-CGI-58 (Proteintech lab, #12201–1-AP, 1:1000 dilution); Rabbit anti-4 Hydroxynonenal (4 HNE) (Abcam, #ab46545, 1:300 dilution); Alexa Fluor 568 anti-Mouse Secondary Antibody (Invitrogen, #A-11004, 1:1000 dilution); Alexa Fluor 633 anti-Rabbit Secondary Antibody (Invitrogen, #A-21070, 1:1000 dilution); Horseradish peroxidase-linked secondary antibodies were purchased from Jackson Immuno-Research Laboratories(1:5000 dilution).[9, 10 (n)-$^3$ hr] triolein, [9, 10 (n)-$^3$ hr] oleic acid were

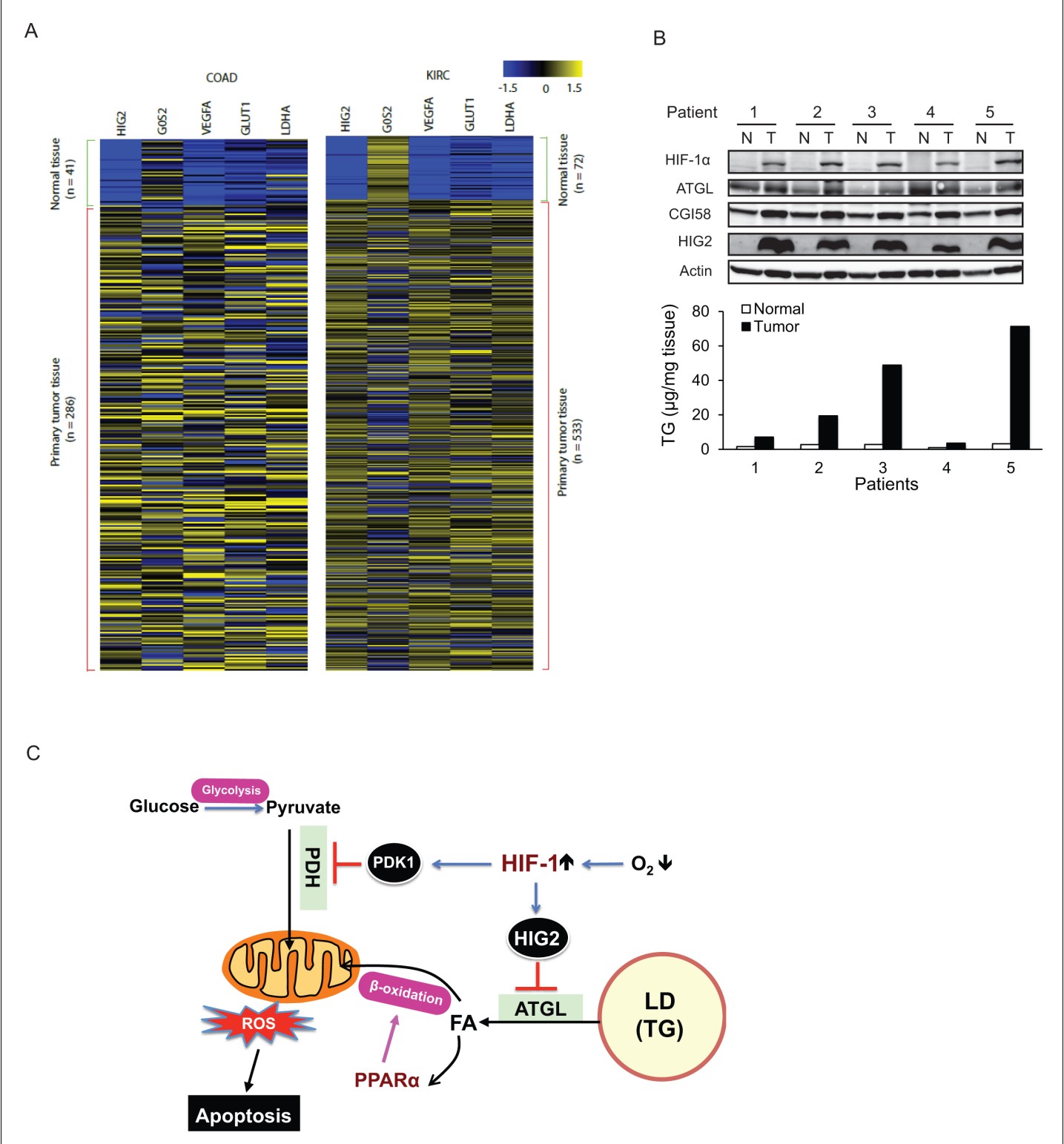

**Figure 8.** The antilipolytic signal is upregulated in human cancers. (**A**) Heat map of gene expression in colon adenocarcinoma (COAD) and kidney renal clear cell carcinoma (KIRC). Pan-cancer normalized expression scores were further Z-score normalized, and fold changes of expression were reported as Log2. *Vegfa*, vascular endothelial growth factor A; *Glut1*, glucose transporter 1; *Ldha*, lactate dehydrogenase A. (**B**) Protein expression and TG content were examined in kidney tissues from human with renal cell cancer. T = kidney tumor, N = adjacent normal kidney tissue. (**C**) The proposed model illustrating the protective role of HIF-1-dependent HIG2 expression and lipid storage against oxidative stress under hypoxia.

DOI: https://doi.org/10.7554/eLife.31132.016

*Figure 8 continued on next page*

*Figure 8 continued*

The following figure supplements are available for figure 8:

**Figure supplement 1.** Differential expression of HIG2 mRNA in tumor specimens and adjacent normal tissues.
DOI: https://doi.org/10.7554/eLife.31132.017
**Figure supplement 2.** HIG2 expression is upregulated in human renal cell cancer tissues.
DOI: https://doi.org/10.7554/eLife.31132.018

from Perkin Elmer Life sciences. Sodium oleate and Oil Red O were from Sigma-Aldrich. cOmplete Mini EDTA-free tablets were from Roche Diagnostics. Lipofectamine RNAiMAX, Lipofectamine 2000, Bodipy 493/503, trypan blue dye and Dead Cell Apoptosis Kits were from Invitrogen. L-Lactate assay kits were from Eton Bioscientific. GW6471 was from Tocris; Ranolazine and trimetazidine were from Cayman Chemical; FA assay kits were from Wako Diagnostics; TG and glucose assay kits were from Thermo Scientific. ImmPACT AMEC Red and Universal Elite ABC kit were from Vector Laboratories.

## Cell culture

HeLa cells were cultured in DMEM (Invitrogen) containing 10% heat-inactivated FBS (Invitrogen). HCT116, DLD-1 and Caki-1 cell were cultured in McCoy's 5A medium (Invitrogen) containing 10% FBS. ACHN cells were cultured in EMEM (ATCC) with 10% FBS. All media were also supplemented with 100 U $ml^{-1}$ penicillin/streptomycin (Invitrogen). Normoxic cells (20% $O_2$) were maintained at 37°C in a 5% $CO_2$/95% air incubator. For hypoxic exposure, cell culture plates were placed in a hypoxia incubator (Eppendorf, USA) at 0.5% $O_2$. All cell lines were obtained from American Type Culture Collection (ATCC). None of the cell lines used was found in the database of commonly misidentified cell lines that is maintained by ICLAC and NCBI Biosample. All cell lines were authenticated by STR profiling and tested to show no mycoplasma contamination.

## PCR cloning of cDNA and site-directed mutagenesis

The complete open reading frame of human HIG2, human or mouse ATGL was cloned into pRK vector without any tags, pKF vector with a FLAG epitope tag, or pKM vector with a Myc epitope tag as described before (*Yang et al., 2010*). Deletion mutations were generated by using the QuickChange site-directed mutagenesis kit (Agilent) according to manufacturer's guidelines.

## RNA extraction and real-time PCR

Total RNA was isolated using the RNeasy Plus Mini Kit (Qiagen) according to the manufacturer's instruction. cDNA was synthesized from total RNA by Omniscript RT Kit (Qiagen). The resulting cDNA was subjected to real-time PCR analysis with iTaq Universal SYBR Green supermix (Bio-Rad) on an Applied Biosystems 7900 HT Real-Time PCR System. The following primers were used: Ribosomal 18S:forward, 5′- GATGGTAGTCGCCGTGCC-3′, reverse, 5′-GCCTGCTGCCTTCCTTGG-3′; ACAA2: forward, 5′- CATGCTGATCTGTTAATGATACCC-3′, reverse, 5′- TGCGTTTTGGAAC-CAAGC-3′; CPT1A: forward, 5′- ATCAATCGGACTCTGGAAACGG-3′, reverse, 5′- TCAGGGAG TAGCGCATGGT-3′; CPT1B: forward, 5′- CCTGCTACATGGCAACTGCTA-3′, reverse, 5′- AGAGG TGCCCAATGATGGGA-3′; MCAD: forward, 5′- GGAAGCAGATACCCCAGGAAT-3′, reverse, 5′-AGCTCCGTCACCAATTAAAACAT-3′; PPARα: forward, 5′- CAGCTCTAGCATGGCCTTTT-3′, reverse, 5′- CCGCAATGGACCATGTAAC-3′; VLCAD: forward, 5′- TCAGAGCATCGGTTTCAAAGG-3′, reverse, 5′- AGGGCTCGGTTAGACAGAAAG-3′. Data were analyzed using the comparative cycle threshold ($^{\Delta\Delta}Ct$) method. The mRNA levels of genes were normalized to Ribosomal 18S. PCR product specificity was verified by post amplification melting curve analysis and by running products on an agarose gel.

## In vitro transcription/translation expression

In vitro transcription/translation was carried out by using TNT SP6 High-Yield Protein Expression System (Promega) according to the manufacturer's instructions. Specifically, reactions consisting of 30 μLTNT SP6 High-Yield Wheat Germ Master Mix and 5 μg vector DNA, made up to 50 μL with molecular biology grade water were incubated for 120 min at 25°C. Then the reaction mixture was used for TG lipase activity.

## Production and purification of bacterially expressed proteins

The human HIG2 or HIG2 7–11 was subcloned by standard PCR into pET His6 MBP vector (addgene, #29708) producing fusion protein with a His6-MBP tag at the Nterminal end. MBP tag can promote the solubility of target protein and His tag enables us easily purify the fusion protein by NI-NTA agarose beads. Plasmids were transformed into BL21 (DE3) E. coli (Agilent), grown in LB media while shaking at 37°C to an OD600 value of 0.6. Protein expression was then induced by adding 1 mM IPTG (Sigma) to the LB cultures, which were shaking at 27°C for another 4 hr. The cells were lysed by sonication/collagenase, and the purification was performed using NI-NTA agrose beads (Qiagen) according to the commercial protocol. The purified protein was then dialyzed overnight in the buffer containing 10 mM Tris-HCl, pH. 7.4, 150 mM NaCl and 1 mM EDTA, aliquoted and stored in −80°C ready for use.

CRISPR/Cas9-mediated gene deletion pSpCas9 (BB)−2A-Puro (pX459) V2.0 was a gift from Feng Zhang (Addgene plasmid #62988). Insert oligonucleotides that include a gRNA sequence were designed using http://crispr.mit.edu/ as follows: for HIG2 deletion, guide1- GGGTCAGTACCA-CACCTAAC, and guide2- GTGTTGAACCTCTACCTGTT; for ATGL deletion, guide- GACCCCGG TGACCAGCGCCG. For co-deletion of HIG2 and ATGL, HIG2 guide one and ATGL guide were used. Cells were seeded in 6-well plates and the following day transfected with pX459 plasmids containing DNA specific to HIG2 and/or ATGL using Lipofectamine 2000. Cells were selected under puromycin (1.5 μg/ml) for 48 hr and plated onto 96-well plates. Screening for genetic modifications was performed by immunoblotting analysis. Mutations were confirmed by direct sequencing. HIG2 deficient clones derived from guide1 were used for experiments unless otherwise indicated.

## Immunoprecipitation analysis

Cells were lysed in cell lysis buffer containing 50 mM Tris-HCl, pH 7.4, 150 mM NaCl, 1% Triton X-100, 1 mM DTT, and protease tablet inhibitors (1 tablet per 10 ml of buffer). Anti-Myc or anti-Flag M2-conjugated agarose gels were incubated with the lysates for 4 hr at 4°C. The beads were then washed four times with lysis buffer, and the bound proteins were eluted in SDS buffer and analyzed by immunoblotting or mass spectrometry.

## Immunoblotting analysis

Cells were lysed at 4°C in a buffer containing 50 mM Tris-HCl (pH 7.4), 150 mM NaCl, 10 mM NaF, 1% Nonidet P-40, 0.1% SDS, 0.5% sodium deoxycholate, 1.0 mM EDTA, 10% glycerol, and protease tablet inhibitors (1 tablet per 10 ml of buffer). The lysates were clarified by centrifugation at 20,000 × g, 4°C for 10 min and then mixed with equal volume of 2 × SDS sample buffer. Equivalent amounts of protein were resolved by SDSPAGE and transferred to nitrocellulose membranes. Individual proteins were blotted with primary antibodies at appropriate dilutions. Peroxide-conjugated secondary antibodies were incubated with the membrane at a dilution of 1:5000. The signals were then visualized using ECL substrate (Thermo Scientific).

## Proteomic analysis

The immunoprecipitated samples were resolved by 10–20% SDS-PAGE gels and visualized by Coomassie Blue staining. Then, the gel portions were excised, de-stained, dehydrated, dried, and subjected to trypsin digestion. The resulting peptides were subjected to liquid chromatography (LC)-ESI-MS/MS analysis performed on a Thermo Scientific LTQ Orbitrap mass spectrometer at the Mayo Clinic Proteomics Core.

## Immunofluorescence microscopy

Cells were seeded on glass coverslips placed in 6-well plates and transfected with 0.25 μg of each DNA construct using Lipofectamine 2000 according to the manufacturer's instructions. Six hours later, transfected cells were incubated with 200 μM oleic acid/0.2% BSA overnight. Following the fixation with 3% paraformaldehyde in PBS for 20 min, cells were permeabilized by 0.5% triton X-100 for 5 min, quenched with 100 mM glycine in PBS for 20 min, and then blocked with 1% BSA in PBS for 1 hr. The cells were then exposed to primary antibody for 2 hr at room temperature. Following three washes with PBS, the cells were treated for 1 hr with Alexa Fluor secondary antibodies. To visualize lipid droplets, 1 μg/ml of Bodipy 493/503 dye was added during the incubation with

secondary antibodies. Samples were mounted on glass slides with Vecta shield mounting medium and examined under a Zeiss LSM 510 inverted confocal microscope. Acquired images were processed and quantified manually with ImageJ FIJI software.

## Assay for TG hydrolase activity

HeLa cells were transfected with ATGL-expressing plasmids using Lipofectamine 2000 overnight and lysed on ice by sonication in a lysis buffer (0.25 M sucrose, 1 mM EDTA, 1 mM Tris-HCl pH 7.4, 1 mM DTT, 20 μg/mL leupeptin, 2 μg/mL antipain and 1 μg/mL pepstatine). The cell extract was clarified by centrifugation at 15,000 g for 10 min, and the supernatant was used as the enzyme source for the assay of TG hydrolase activity. The TG lipase activity was determined using a lipid emulsion labeled with [9,10-$^3$H]- triolein as substrate. For HIG2 obtained from HeLa cells transfected with HIG2- expressing plasmids, 50 μl of HIG2 lysate was combined with 30 μl of ATGL lysate; For HIG2 derived from In vitro transcription/translation expression, 25 μl of In vitro transcription/translation reaction was combined with 25 μl of lysate buffer and 30 μl of ATGL lysate; For HIG2 purified from E. coli, 1 μg of protein diluted in 50 μl of lysate buffer and was combined with 30 μl of ATGL lysate. Then the 80 μl HIG2/ATGL mixture was incubated with 80 μl of substrate solution for 60 min at 37°C. Reactions were terminated by adding 2.6 ml of methanol/chloroform/heptane (10:9:7) and 0.84 mL of 0.1 M potassium carbonate, 0.1 M boric acid (pH 10.5). Following centrifugation at 800 × g for 15 min, radiolabeled fatty acids in 1 ml of upper phase were measured by liquid scintillation counting.

## Lipolysis assay and measurement of cellular TG content

Lipolysis was measured as the rate of free fatty acid release. In brief, cells were cultured under normoxia or hypoxia with 10% FBS medium in the presence of 200 μM oleate/0.2% BSA complex. Twenty four hours later, cells were washed twice with PBS and incubated with serum-free medium without phenol red containing 1% BSA for another four hours. Then medium was collected and FAs released were determined by a FA assay kit according to the manufacturer's instructions. Lysates were then prepared from the remaining cells, and protein concentrations in the lysates were used to normalize FFA levels in the medium. The relative rate of lipolysis (%) was calculated based on the relative concentration of FFAs among the groups. For TG measurement, cells or human samples were lysed in lysis buffer (1% NP-40, 150 mM NaCl, 10 mM Tris-HCl, pH7.4). Equal volume of cell lysates were used to measure TGs using a triglyceride assay kit according to the manufacturer's instructions. TG concentration was calculated and normalized to protein contents.

## HIF-1α knockdown by siRNA

Cells were seeded at 20–40% confluency and the day after transfected with 25 nM siRNA using Lipofectamine RNAiMAX according to the manufacturer's protocol. One day later, cells were incubated under normoxia or hypoxia with fresh McCoy's 5A Medium containing 10% FBS and processed for designated assays. The following stealth siRNA oligonucleotides (Invitrogen) were used for human HIF1α knockdown: 5′- GGGAUUAACUCAGUUUGAACUAACU-3′ (sense) and 5′- AGUUAG UUCAAACUGAGUUAAUCCC-3′. Control oligonucleotides with comparable GC content were also purchased from Invitrogen.

## Fatty acid oxidation measurement

Fatty acid oxidation was assessed on the basis of $^3$H$_2$O production from [9,10- $^3$H]-oleate as previously described with minor modifications (*Zhang et al., 2014*) unless otherwise stated. Cells in 6-well plates were washed with PBS, and then incubated with 2 ml of BSA-complexed oleate (0.2 mM unlabeled plus 2 μCi/ml of [9, 10- $^3$H] oleate and 0.2% BSA) in serum-free medium. Six hours later, the medium was collected for measuring $^3$H$_2$O production. Briefly, excess [9,10- $^3$H]- oleate in the medium was removed by precipitating twice with an equal volume of 10% trichloroacetic acid with 2% BSA. After centrifugation at 15,000 × g for 3 min at 4°C, the supernatants (0.5 ml) were extracted with 2.5 ml of chloroform/methanol (2:1, v/v) and 1 ml of 2 M KCl/HCl (1:1, v/v), following by centrifuged at 3,000 × g for 5 min. The aqueous phase containing $^3$H$_2$O was collected and subjected to liquid scintillation counting and data was normalized by protein contents.

## ROS measurement and cell apoptosis assay by flow cytometry

For ROS measurements, cells were washed with PBS and stained with 2.5 μM $H_2DCFDA$ in PBS at 37°C for 30 min. Then cells were trypsinized, washed with PBS and re-suspended in PBS. Stained cells were filtered and DCF fluorescence was measured using a FACSCelesta flow cytometer (BD Biosciences) and FACSDiva software. For apoptosis detections, cells were harvested and washed with PBS. Pellets were resuspended in 1X Annexin binding Buffer, and stained with Alexa Fluor 488 annexin V and PI for 15 min at room temperature. Stained cells were filtered and analyzed immediately by flow cytometer. Apoptosis data are presented as the percent fluorescence intensity of gated cells positive for Annexin V.

## Quantification of viable cells

$2.5 \times 10^5$ cells were planted in 12 well plates 1 day before exposure to hypoxic conditions (0.5% $O_2$). At the times indicated, cells were trypsinized and the viable cells were counted using trypan blue dye exclusion test.

## Lactate production and glucose consumption assay

Cells were cultured under normoxia or hypoxia for 24 hr. Glucose and lactate levels in the culture medium were determined using a glucose assay kit and a lactate assay kit, respectively, according to manufacturer's instructions. Data were normalized to protein contents.

## Immunohistochemistry

Xenograft tumors were fixed in 4% paraformaldehyde and embedded in paraffin. Sections were deparaffinized by baking slides at 55°C for 15 min, rehydrated in xylene and series of ethanol solutions, and endogenous peroxidase activities were quenched by 3% $H_2O_2$ for 10 min. Antigen retrieval was performed by pepsin digestion for 10 min and sections were blocked for 30 min in 2.5% normal horse serum. Then sections were incubated with anti-cleaved Caspase-3 or anti-4 HNE antibody (1:200 dilution) overnight at 4°C. Sections were further incubated for 40 min with Imm-PRESS-AP Anti-Rabbit IgG Reagent (Vector Laboratories), and then processed with Permanent Red Substrate-Chromogen (Agilent Technologies) and Methyl Green solutions (Agilent Technologies) for immunohistochemistry staining. Sections were photographed at ×20 magnification, and the staining intensity was measured by using ImageJ FIJI software.

## Oil red O staining

Frozen sections from xenograft tumors were fixed in 10% formalin, washed with propylene glycol, and stained with 0.5% oil red O in propylene glycol for 10 min at 60°C. Then the slides were differentiated in 85% propylene glycol, washed with water, and counterstained with Mayer's Hematoxylin. The red lipid droplets were visualized by microscopy.

## Mouse xenografts

Male athymic nude mice, 6–7 weeks of age, were purchased from Taconic Biosciences. The experimental procedures were approved by the Mayo Clinic Institutional Animal Care and Use Committee. Animals arriving in our facility were randomly placed in cages with five mice each. They were implanted with respective tumor cells in the unit of cages, which were randomly selected. Cells were prepared in in phenol red-free culture medium without FBS. Two hundreds microliter of $3 \times 10^6$ HCT116 clone cells or $2 \times 10^6$ HeLa clone cells were injected subcutaneously into both flanks of nude mice. Tumor size was measured twice weekly by calipering in two dimensions and the tumor volume in mm3 is calculated by the formula: $(width)^2$ x length/2. After 25 days, mice were killed, and tumors were dissected and weighed.

## TCGA data analysis

TCGA RNA-seq expression data were downloaded from the UCSC (University of California, Santa Cruz) cancer browser (https://genome-cancer.ucsc.edu/), in which the gene expression values were measured by log2 transformed, RSEM normalized read counts (*Cline et al., 2013*; *Li and Dewey, 2011*). Differential gene expression analyses between tumor and matched normal tissue for HIG2

were performed in 20 cancer types that have at least two available normal tissues. *P* values were evaluated using the Wilcoxon rank-sum test (i.e. Mann-Whitney U test).

## Human tissue biospecimens

Frozen biospecimens were collected as previously described (*Ho et al., 2015*). The biobank protocol was approved by the Mayo Clinic Institutional Review Board (protocol no. 08–000980) and patient informed consent was obtained from all subjects. After review by a genitourinary pathologist, frozen matched tumors and adjacent uninvolved kidney were selected for further study. Criteria for selection included tumor sample composed of viable-appearing tumor cells with $\geq$60% tumor nuclei and $\leq$20% necrosis of sample volume.

## Statistics and reproducibility

Sample sizes and statistical tests for each experiment are denoted in the figure legends or in the figures. Data analysis were performed using Excel software (2013) and values are expressed as mean ±SD or SEM. Values are expressed as mean ±SD or SEM. The unpaired two-tailed Student's t-test was used to determine the statistical significance of differences between means ($p<0.05$, $p<0.01$, $p<0.001$) unless otherwise indicated. All experiments were repeated independently at least three times with similar results, except for Mass Spectrometry, patient samples and animal experiments shown in *Figure 2a,b*, *Figure 7a,b,c,d*, *Figure 8b* and supplementary *Figure 5a,b*. There is no estimate of variation in each group of data and the variance is similar between the groups. No statistical method was used to predetermine sample size. The investigators were not blinded to allocation during experiments and outcome assessment. All data were expected to have normal distribution. None of the samples/animals was excluded from the experiment.

## Acknowledgements

This work was supported by research grants from the National Institutes of Health (DK089178 and DK109096) to JL, and from the Office of the Assistant Secretary of Defense for Health Affairs (W81XWH-17-1-0546) to T.H.H.

## Additional information

### Funding

| Funder | Grant reference number | Author |
| --- | --- | --- |
| U.S. Department of Defense | W81XWH-17-1-0546 | Thai H Ho |
| National Institute of Diabetes and Digestive and Kidney Diseases | DK089178 | Jun Liu |
| National Institute of Diabetes and Digestive and Kidney Diseases | DK109096 | Jun Liu |

The funder had no role in study design, data collection and interpretation, or the decision to submit the work for publication.

### Author contributions

Xiaodong Zhang, Conceptualization, Data curation, Formal analysis, Validation, Investigation, Visualization, Methodology, Writing—original draft, Writing—review and editing; Alicia M Saarinen, Data curation, Validation, Investigation, Visualization, Methodology; Taro Hitosugi, Data curation, Formal analysis, Investigation, Methodology, Writing—review and editing; Zhenghe Wang, Data curation, Methodology, Writing—review and editing; Liguo Wang, Data curation, Software, Formal analysis, Methodology, Writing—review and editing; Thai H Ho, Resources, Data curation, Investigation, Methodology, Writing—review and editing; Jun Liu, Conceptualization, Resources, Data curation, Formal analysis, Supervision, Funding acquisition, Investigation, Methodology, Writing—original draft, Project administration, Writing—review and editing

**Author ORCIDs**

Jun Liu http://orcid.org/0000-0002-3646-0004

**Ethics**

Animal experimentation: This study used male athymic nude mice purchased from Taconic Biosciences. All of the animal experimental procedures were approved by the Mayo Clinic Institutional Animal Care and Use Committee. (IACUC Protocol A00001813-16).

**Decision letter and Author response**

Decision letter https://doi.org/10.7554/eLife.31132.021
Author response https://doi.org/10.7554/eLife.31132.022

## Additional files

**Supplementary files**
• Transparent reporting form
DOI: https://doi.org/10.7554/eLife.31132.019

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
