## [Decision Letter]

Thank you for submitting your article "Inhibition of Intracellular Lipolysis Promotes Cancer Cell Adaptation to Hypoxia" for consideration by *eLife*. Your article has been favorably evaluated by Mark McCarthy (Senior Editor) and three reviewers, one of whom, Ralph DeBerardinis (Reviewer #1), is a member of our Board of Reviewing Editors.

The reviewers have discussed the reviews with one another and the Reviewing Editor has drafted this decision to help you prepare a revised submission.

Summary:

This paper studies the role of HIG2 in lipid droplet (LD) dynamics and cell survival during hypoxia.. Proteomics and functional studies demonstrate that a HIF-1 target, hypoxia inducible gene-2 (HIG2) is a novel inhibitor of adipose triglyceride lipase (ATGL). During hypoxia, HIG2 binds to ATGL, suppresses its lipase activity and prevents it from catabolizing LDs. During prolonged hypoxia, HIG2's association with ATGL is required for cell survival, and the authors present evidence that the mechanism of cell death under HIG2-silenced conditions involves reactive oxygen species arising from unconstrained fatty acid oxidation (FAO). In xenografts, HIG2 deletion reduces lipid deposition and suppresses tumor growth, but these effects are rescued by co-deleting ATGL. The authors also present evidence that HIG2 expression tends to be enhanced in human cancers from several organ sites. Overall this is an interesting and clearly written paper with conclusions supported by the data. A few additional experiments and analyses would strengthen the conclusions.

Essential revisions:

1) The authors' model indicates that fatty acids released from LDs are oxidized to produce ROS. Although the data indicate that HIG2 is required for FAO and ROS production, two additional experiments should be performed. First, the authors should formally test whether inhibiting FAO mitigates ROS and cell death during HIG2 silencing. Second, the FAO assay uses exogenous fatty acids as a source of tracer. It is not clear that these fatty acids are first sent to the LD before being oxidized, and given the effects of HIG2 on expression of FAO enzymes, it is possible that the major substrate for FAO is the exogenous fatty acid pool rather than LD hydrolysis. It should be possible to load the LD pool with labeled fatty acids, then wash out the label, subject the cells to hypoxia and measure FAO. If the liberation of labeled water is still enhanced by HIG2 knockout, this would provide additional support to the model in Figure 8.

2) The patient data is preliminary. A large number of patient materials are characterized as having elevated HIG2 mRNA expression in Figure 8. However, protein data in Figure 8 only depicts 5 patient samples. A larger cohort of human patient material tissue sections should be acquired to show that primary patient tissues accumulate HIG2 protein, and ideally that this is also inversely correlated with markers of lipid peroxidation, such as by 4-HNE immunostaining. This would increase the conclusions drawn from the paper and further bolster the model shown in Figure 8.

3) Most of the data in Figure 7 are convincing, but the quality of the lipid peroxidation and cleaved Caspase 3 IHCs in Figure 7 is poor. These data should be improved and quantified to show convincing reproducibility in the distinct experimental cohorts.

4) Some formal quantification of LD staining, at least for key experiments, should be added to complement the images.

---

## [Author Response]

Essential revisions:1) The authors' model indicates that fatty acids released from LDs are oxidized to produce ROS. Although the data indicate that HIG2 is required for FAO and ROS production, two additional experiments should be performed. First, the authors should formally test whether inhibiting FAO mitigates ROS and cell death during HIG2 silencing. Second, the FAO assay uses exogenous fatty acids as a source of tracer. It is not clear that these fatty acids are first sent to the LD before being oxidized, and given the effects of HIG2 on expression of FAO enzymes, it is possible that the major substrate for FAO is the exogenous fatty acid pool rather than LD hydrolysis. It should be possible to load the LD pool with labeled fatty acids, then wash out the label, subject the cells to hypoxia and measure FAO. If the liberation of labeled water is still enhanced by HIG2 knockout, this would provide additional support to the model in Figure 8.

We have performed both experiments as suggested. As shown in the new Figure 6 and Figure 6—figure supplement 3, treatment with FAO inhibitor such as Ranolazine or Trimetazidine both attenuated apoptotic induction by hypoxia in HIG2 KO cells. Also, ROS levels were considerably reduced in Ranolzine-treated cells under hypoxia (Figure 6). Moreover, we assayed FAO in normoxia or hypoxia after we prelabeled the intracellular TG pool with ^3^H-oleate in normoxia. The data presented in the new Figure 6—figure supplement 3 showed increased FAO in the hypoxic HIG2 KO cells, which are consistent with the results presented in Figure 6 and with our model depicted in Figure 8.

2) The patient data is preliminary. A large number of patient materials are characterized as having elevated HIG2 mRNA expression in Figure 8. However, protein data in Figure 8 only depicts 5 patient samples. A larger cohort of human patient material tissue sections should be acquired to show that primary patient tissues accumulate HIG2 protein, and ideally that this is also inversely correlated with markers of lipid peroxidation, such as by 4-HNE immunostaining. This would increase the conclusions drawn from the paper and further bolster the model shown in Figure 8.

We have obtained 14 more pairs of frozen human kidney tissue samples from the Mayo Clinic Biobank. As shown in the new Figure 8—figure supplement 2, HIG2 and HIF-1alpha proteins were detected only in cancer tissues but not in the paired normal tissues. The total TG content followed a similar trend. Unfortunately, we were unable to perform co-staining of 4-HNE and HIG2 protein since the antibody is not suitable for immunostaining of endogenous protein in tissue sections. However, given the universal upregulation of HIG2 protein in all 19 RCC samples examined (Figure 8 and Figure 8—figure supplement 2) and the bioinformatics analysis revealing a strong association of the mRNA abundance of HIG2 and other HIF-1 target genes with multiple solid tumors including RCC and CRC (Figure 8 and Figure 8—figure supplement 1), we feel that the evidence in support of tumor-specific expression of HIG2 in human patients is compelling.

3) Most of the data in Figure 7 are convincing, but the quality of the lipid peroxidation and cleaved Caspase 3 IHCs in Figure 7 is poor. These data should be improved and quantified to show convincing reproducibility in the distinct experimental cohorts.

We have repeated the IHCs with a new colorimetric reagent, which produced a substantially reduced background. The new images with quantification are now included in Figure 7 and Figure 7—figure supplement 1.

4) Some formal quantification of LD staining, at least for key experiments, should be added to complement the images.

We have included quantification of LD staining in Figure 4. The data are consistent with the biochemical measurement of the total cellular TG levels as presented in Figure 4.